# Manufacturing of All Inkjet-Printed Organic Photovoltaic Cell Arrays and Evaluating Their Suitability for Flexible Electronics

**DOI:** 10.3390/mi9120642

**Published:** 2018-12-04

**Authors:** Kalyan Yoti Mitra, Abdelrahman Alalawe, Stefanie Voigt, Christine Boeffel, Reinhard R. Baumann

**Affiliations:** 1Fraunhofer Institute for Electronic Nanosystems ENAS, Printed Functionalities, 09126 Chemnitz, Germany; reinhard.baumann@enas.fraunhofer.de; 2Digital Printing and Imaging Technology Department, Technische Universität Chemnitz, 09126 Chemnitz, Germany; reinhard.baumann@mb.tu-chemnitz.de; 3Fraunhofer Institute for Applied Polymer Research, Functional Materials and Devices, 14476 Potsdam, Germany; stefanie.voigt@iap.fraunhofer.de (S.V.); christine.boeffel@iap.fraunhofer.de (C.B.)

**Keywords:** organic photovoltaics, flexible electronics, Indium Tin Oxide (ITO) free solar cells, inkjet technology

## Abstract

The generation of electrical energy depending on renewable sources is rapidly growing and gaining serious attention due to its green sustainability. With fewer adverse impacts on the environment, the sun is considered as a nearly infinite source of renewable energy in the production of electrical energy using photovoltaic devices. On the other end, organic photovoltaic (OPV) is the class of solar cells that offers several advantages such as mechanical flexibility, solution processability, environmental friendliness, and being lightweight. In this research, we demonstrate the manufacturing route for printed OPV device arrays based on conventional architecture and using inkjet printing technology over an industrial platform. Inkjet technology is presently considered to be one of the most matured digital manufacturing technologies because it offers inherent additive nature and last stage customization flexibility (if the main goal is to obtain custom design devices). In this research paper, commercially available electronically functional inks were carefully selected and then implemented to show the importance of compatibility between OPV material stacks and the device architecture. One of the main outcomes of this work is that the manufacturing of the OPV devices was accomplished using inkjet technology in massive numbers ranging up to 1500 containing different device sizes, all of which were deposited on a flexible polymeric film and under normal atmospheric conditions. In this investigation, it was found that with a set of correct functional materials and architecture, a manufacturing yield of more than 85% could be accomplished, which would reflect high manufacturing repeatability, deposition accuracy, and processability of the inkjet technology.

## 1. Introduction

Inkjet technology is a powerful noncontact digital deposition technique with accuracy in the micrometer range. It deposits numerous numbers of functional materials onto various categories of substrates and surfaces. It is also an economical manufacturing technique that uses less material than any other traditional printing methods [1]. The technology offers precise deposition of materials, as the ink drops are only jetted when it is required. These drops are then guided in a specified sequence and arrangement, creating the image on the substrate. Inkjet technology offers freedom of shape and design on an office, laboratory, and industrial scale. The ink gets transported directly from the printhead to the substrate, as it does not require a master, plate, or carrier to transfer the printing image. The technology furthermore supports the digital manufacturing of devices, allowing a late-stage customization of series manufacturing, as demanded by the consumer market. In addition to this, it is widely used in the production of several printed electronic devices (e.g., radio-frequency identification (RFID) antennas [2,3,4], capacitors [5,6], low pass filters [5,6], organic thin film transistors [7,8,9,10,11,12], organic photovoltaics (OPVs) [13,14,15], electrical components on different materials [6], pressure sensors [16], batteries [17,18,19,20], organic light emitting diodes [21,22,23,24,25,26,27], and the capability to manufacture further devices). Thus, inkjet technology can be considered as a matured technology for an accurate deposition process of electronically functional ink materials.

This research aims for the first time to manufacture all inkjet-printed indium tin oxide (ITO)-free organic photovoltaics (OPVs) under ambient conditions over a sheet-to-sheet (S2S)-based industrial scale and using a polymeric flexible polyethylene naphthalate (PEN) substrate. The OPVs based on the conventional architecture were composed of a layer stack i.e., a bottom cathode electrode, an electron transport layer (ETL), a charge donor-acceptor layer (photoactive layer), a hole transport layer (HTL), and a top anode electrode. The field of photovoltaics is generally categorized into different types e.g., thin-film solar cells, dye-sensitized solar cells, tandem solar cells, and bulk heterojunction (BHJ) based organic solar cells [28,29,30]. Nevertheless, the category that supports printing technology in particular is the solution-based OPVs that utilize BHJ as the photoactive functional layer, and thus the focus of this research also was on the manufacturing of OPVs by implementing BHJ. The chosen BHJ material was a blend of polymer to fullerene, polymer-based poly-3-hexyl thiophene (P3HT) conjugated polymers donor-blended with an organic fullerene acceptor, (6,6)-phenyl-C60-butyric acid methyl ester (PCBM). The P3HT:PCBM material was of choice due to its broadband implementation into the manufacturing of printed OPVs and its air-stable characteristics. This category of semiconducting materials is comparatively cheap, flexible, lightweight, and solution-processable into various ratios or concentrations, which allows the deposition of this material using inkjet technology on a large printable scale. The manufacturing of an OPV stack requires generally the deposition of several materials (e.g., zinc, silver, or aluminum) for the deposition of cathode electrodes. Meanwhile, the transparent anode electrode could fundamentally be obtained using ITO or poly(3,4-ethylenedioxythiophene) polystyrene sulfonate (PEDOT:PSS). The deposition of the photoactive layer was performed using a variety of materials such as the conjugated polymer, donor material poly-3-hexyl thiophene (P3HT), or poly (*N*-9′-heptadecanyl-2,7-carbazole-alt-5,5-(4′,7′-di-2-thienyl-2′,1′,3′benzothiadiazole)) (PCDTBT). Furthermore, fullerene derivatives such as acceptor material phenyl-C_60_-butyric acid methyl ester (PC_60_BM), PC_70_BM, PC_84_PM, and endohedral fullerenes are some alternative options [31]. Moreover, for the HTL layer tungsten oxide (WO_3_) and poly (3,4-ethylenedioxythiophene) polystyrene sulfonate (PEDOT:PSS) are implemented, and for the ETL layer materials such as zinc oxide (ZnO), aluminum-doped zinc oxide (AZO), and tin oxide (SnO_X_) are generally implemented [31]. Indeed, most of these materials are commercially available and could be optimized for the inkjet printing applications [32]. Numerous research groups have involved themselves in the development of OPVs using several deposition technologies, including printing technologies such as screen printing, gravure printing, inkjet printing, spin coating, and thermal evaporation [33,34,35,36,37,38,39,40,41]. For instance, a research group reported about all screen-printed OPV stacks [34]. In fact, the stack was based on the use of expensive ITO-coated polyethylene terephthalate (PET) substrate as the bottom transparent electrode on a laboratory (Lab) scale as well as on a large area industrial fabrication scale (Fab) [34]. Another group implemented gravure printing technology in the production line of only a few layers for the OPVs, HTL and the photoactive layer, whereas the rest of the layers were obtained using further methods such as thermal evaporation of the top electrode [35,36]. In the field related to the implementation of inkjet technology, other articles have reported about the printing of OPVs using a single nozzle homebuilt Lab inkjet printer using a piezoelectric MJ-A system, from Microfab printhead (MicroFab Technologies, Inc., Plano, TX, USA). The device structure of the inkjet-printed devices was deposited on a glass substrate with the following sequence: ITO/PEDOT:PSS/PCDTBT:PC_70_BM/ZnO/Ag. PEDOT:PSS (HTL) was printed with thicknesses from 110 nm to 200 nm, and dried at 120 °C for 15 min. The photoactive layer was deposited using PCDTBT:PC_70_BM and cured at 70 °C for 10 min. The nanoparticle-based ZnO (ETL) and Ag cathode inks were sintered at 80 °C for 15 min. It is worth mentioning that both the organic donor and acceptor materials were found to be very sensitive to temperature, humidity, and high costs. The active area of a single device was about 0.5 cm^2^, and the performance of the devices was evaluated in ambient condition with an Air Mass 1.5 Global solar simulator (Solar Light Company, Inc., Glenside, PA, USA) with an irradiation intensity of 100 mW/cm^2^. The voltage versus current output characteristics of the devices were measured using the Keithley 236 Source Measurement Unit. The average power conversion efficiency (PCE) of the all inkjet-printed devices was about 2%. These devices were printed only with the Lab inkjet printer on the expensive ITO-based PET substrate. Moreover, devices were further fabricated using a mixture of solvents for the photoactive layer, which led to an enhancement in the OPV performance [14]. Another article has reported about the manufacturing of inkjet-printed OPVs based on the ITO-coated glass substrate, combining both laboratory and industrial equipment for the development of a single stack in “clean room 1000” conditions. A special uncoated glass substrate as well as an ITO-coated glass substrate were used as a basic front electrode and substrate, whereas the back electrode was printed using two different designs separately (i.e., full area and grids). The Ag finger grids were printed using Lab inkjet printer DMP-2831 from Fujifilm Dimatix (Santa Clara, CA, USA) with 16-nozzle DMC printheads. The deposition of remaining layers was done using an industrialized inkjet printhead KM512LN from Konika Minolta (Tokyo, Japan), with a 3.5 cm width and 360 dpi nozzle spacing with 512 nozzles on the LP50 printing platform, from Pixdro (Eindhoven, Netherlands). The sequence of depositing the layer stack using both the inkjet systems was the PEDOT:PSS-Ag finger grid/ZnO/P3HT:PCBM/PEDOT:PSS-Ag finger grid, in clean room conditions. To compare the device performances, stacks were also produced using an alternative BHJ material, commercially available ActiveInk PV2000 semiconductor from Polyera Corporation (Skokie, IL, USA). The thicknesses of the printed layers were measured using Dektak profilometry (Veeco Instruments, Inc., Plainview, NY, USA) and were found to be about 50 nm for the ZnO, 220 nm for P3HT:PCBM, 260 nm for the PV2000-based BHJ, and 200 nm for the PEDOT:PSS layers. The devices were annealed at different temperature and durations, at 130 °C for 10 min for P3HT:PCBM and at 120 °C for 3 min for the PV2000-based devices, all under an N_2_ atmosphere. The current density as well as the voltage values were measured under a halogen lamp (100 mW/cm^2^). The electrical parameters of the full area printed back electrodes were PCE 1.7%, *V_oc_* 0.57 V, *I_sc_* 5.64 mA/cm^2^, and fill factor (FF) 52.4%. Meanwhile, for the OPVs containing grids as back electrodes, the PCE was 1.6%, *V_oc_* was 0.56 V, *I_sc_* was 5.46 mA/cm^2^, and FF was 51.6%. Furthermore, the devices with the ActiveInk PV2000 (Sigma-Aldrich, St. Louis, MO, USA) semiconductor showed even better performance [15]. 

These reports are clear indicators that the inkjet technology is a potential technique that has the capability to develop printed products by manufacturing large-area organic electronics with comparable performance to the traditional techniques. Meanwhile, benefiting from the digital printing process, advantages over industrial scales due to freedom of design, short runs, and printing on demand could also be addressed. However, the number of research projects on all layers printed with inkjet technology for the development of OPVs are first in the minority, and second, there are none that have also shown the tolerances with respect to the device performance, scalability, and manufacturing yield or processing reliability. Therefore, this research paper reports for the first time the escalation of the manufacturing process for ITO-free OPVs over flexible polymeric substrates in the form of OPV arrays and size varied from 5 mm^2^ to 20 mm^2^, completely under ambient conditions. The research will define the various statistical tolerances that were exhibited in the functional regime for the set of functional materials and related manufacturing technologies. It starts from the Lab to Fab scale for all the essential functional inks (e.g., organic BHJ, organic and inorganic nanoparticle-based conductive materials, and charge transport materials) that are fundamentally required to develop functional OPVs. 

## 2. Methods and Materials

### 2.1. Stack Architecture and Patterns

A device architecture based on the conventional stack was selected to print OPVs using inkjet technology. A conventional stack OPV architecture promotes passage of the light illumination through the top electrode (anode) incident into the photoactive (organic BHJ) layer of the device. The intended OPV architecture is developed based on the material’s energy band levels, which has been implemented already in the literature [14,42,43] and is also explained in Appendix A. Figure 1A shows a typical architecture of the OPV device. A simplified and yet systematic approach was adopted to deposit all the functional layers using solid (fully covered) and closed layers, without any implementation of intermediate grids. A much-recognized manufacturing workflow was followed, where the preliminary printing tests were first performed on the DMP-2831 Lab inkjet printer from Fufifilm Dimatix and finally the entire Fab process was upscaled over an industry relevant inkjet printer DMP-3000. During this process, the deposition parameters were transferred from the Lab to the Fab scale. The deposition of the functional material layers in the Lab platform was obtained using 10 pL drop volume (DMC cartridges), whereas the ones in the upscaling platform were accomplished using 35 pL (SE3 printheads). For the sake of convenience, a vector-based graphic editing software, “Adobe Illustrator” from Adobe Systems (Adobe Illustrator CC, Adobe Inc., San Jose, CA, USA), was used to create all the required digital patterns. As shown in Figure 1B, the printed devices were obtained using a specific order i.e., the bottom silver (Ag) cathode electrode layer was printed using a nanoparticle-based ink directly on the flexible plastic film. The layer layouts were designed to address the three different device sizes i.e., 5 mm × 1.5 mm (7.5 mm^2^), 5 mm × 2.5 mm (12.5 mm^2^), and 5 mm × 3.5 mm (17.5 mm^2^). All three digital patterns were generated inside one design, and the pattern was printed using a 40 μm drop space (635 dpi). A Zinc Oxide (ZnO) based electron transport layer (ETL) was printed over the Ag electrode as the second layer. The digital dimensions for the ETL were designed to be larger than the active area of the Ag electrode 7.5 mm × 9 mm (67.5 mm^2^), with the goal of completely covering the electrode’s active area. The corresponding patterns were printed with 20 μm drop space (1270 dpi). The same digital pattern layout was also used for the third layer, which was the photoactive organic bulk-heterojunction (BHJ) layer i.e., blend of poly(3-hexylthiophene) to phenyl-C60-butyric acid methyl ester (P3HT:PCBM) for the OPVs. But, here the patterns were deposited using a 25 μm drop space (1016 dpi). The top anode electrode was deposited on top of the photoactive layer using a polymer-based poly(3,4-ethylenedioxythiophene) polystyrene sulfonate (PEDOT:PSS) conductor. The layer was printed using 15 μm (1693 dpi) and 11 μm (2309 dpi) drop spaces to achieve the required electrode conductivity. The physical dimension of the top electrodes was designed with the same dimension as that of the bottom Ag electrode, but with 90° angular rotation. This reorientation of the pattern was done to prevent any unnecessary interference between the anode and cathode electrodes. 

As mentioned before, the main intention of this research work was to exhibit the process of upscaling via inkjet technology. The primary implementation was done for the target-printed OPVs by applying conventional architecture that was suitable for inkjet printing technology and its industrialization. Here, special focus has been put forward on the process development for the OPV devices—those that are based on ITO free electrodes and all printed functional layers cured under normal atmospheric conditions—without the use of any special environment. With this focus, one of the vital investigations was also to engage in the concept of upscaling, which is increasing the rate of deposition by using a higher number of nozzles i.e., 16 to 128, a drop volume from 10 pL to 35 pL, and industrial equipment for the printing process. Besides this, the device size was escalated, and the correlation between the corresponding OPV performance and size was established. A statistical analysis of the tolerance in the device performance and fabrication yield with respect to the printed device size was also investigated. The main content of the investigation was based on the deposition of OPV devices over a DIN A4-sized substrate, where six arrays containing 216 devices were implemented for each substrate sheet and up to five sheets were considered. For each individual array, there were always 36 devices arranged in symmetric order equally spaced and positioned in the printable area. A schematic representation of the pattern and its layout is shown in Figure 2. In the figure, all four fundamental layers are clearly designated, where the geometrical dimension of the ETL and BHJ layers were kept constant at 7.5 mm × 9 mm (67.5 mm^2^). However, there was a definite difference in the active areas in the addressed six devices, that were located in the six different arrays. The difference in the active area was created by varying the printable area of the bottom and top electrode, which were in general of the same dimension, except for their 90° rotated orientation. The 6 different considered active areas in the present investigation were: Array 1 (5 mm^2^), Array 2 (7.5 mm^2^), Array 3 (10 mm^2^), Array 4 (12.5 mm^2^), Array 5 (17.5 mm^2^), and Array 6 (20 mm^2^). The bottom electrode or cathode was printed using a 65 μm drop space (391 dpi). The second ETL layer was printed twice using a 40 μm drop space (635 dpi), deposited (printed wet on wet and without curing) on top of each other, over the bottom electrode. This was done to ensure accomplishment of homogeneous pin holes-free layers. The BHJ layer was also printed two times over each other (wet on wet and without curing), achieving a homogeneous layer using a 40 μm drop space (635 dpi). Finally, the deposition of the top anode electrode was accomplished by printing 2–3 consecutive layers using a 25 μm drop space (1016 dpi) in order to obtain a homogeneous layer without any artifacts.

### 2.2. Substrate and Functional Inks

All the experiments were conducted on an acrylic-coated polyethylene naphthalate (PEN) Teonex Q65HA film from DuPont Teijin (Chester, VA, USA), with a thickness of 125 μm. It is a highly transparent smooth PEN substrate with superior coated film that makes it suitable for inkjet printing process. It has an excellent property of thermal stability with a maximum processing temperature of up to 180 °C, which offers a thermal shrinkage of 0.1% at 150 °C for 30 min. These properties make the PEN film perfect substrate for printed electronics. In total four inks were implemented to deposit the different layers for the manufacturing of the printed OPVs. The first ink that was used to deposit the bottom cathode electrode on the PEN film was the silverjet DGP LT-15C nanoparticle-based Ag ink from Advanced Nano Products/Sigma Aldrich (St. Louis, MO, USA). It contains 30–35 wt % Ag nanoparticles in the solvent triethylene glycol mono methyl ether. The ink offers an electrical resistivity of 11 µΩ/cm, and it is commercialized especially for printing on plastic films. The second functional layer, the ETL, was deposited using Nanograde N11 Jet, which is a nanoparticle-based ZnO ink from Avantama AG (Stäfa, Switzerland). It is a translucent light brownish colored ink, having a concentration of 2.5 wt % crystalline ZnO in 2-propanol and propylene glycol solvents. The N11-Jet material ink offers a material work function of −3.9 eV, and it is formulated especially for organic electronics using inkjet printing technology [44]. For the deposition of the photoactive BHJ layer, a blend of P3HT:PCBM in a ratio of 1:1 (donor to acceptor) was implemented. Both the organic components were acquired from Osilla Ltd. (Sheffield, UK). and dissolved in the solvent 1,2-dichlorobenzene (oDCB) with a concentration of 12.5 mg/mL (total solid content). The concentration of the material to the solvent was specially optimized to facilitate the inkjet printing process. Finally, the PEDOT:PSS-based semitransparent top anode electrode was deposited, using ink acquired from Heraeus group (Hanau, Germany) with a trademark name of CLEVIOS™ F HC Solar. The ink is dark blue in color and is specially developed for OPV applications. The ink contains a solid content of 1.16 wt % in combination with polar solvents such as water and other surfactants. The ink has a surface tension of about 30 mN/m, which allows good wettability with a contact angle less than 30° on the low energized organic BHJ layer surfaces. 

### 2.3. Deposition Process

The inkjet printer Dimatix Materials Printer DMP-2831 from Fujifilm Dimatix was used to deposit the different layers for the OPV device. Typically, the printer is based on a low-cost piezoelectric actuation platform based on a drop-on-demand “DoD” inkjet system that uses refillable and fairly cost-effective DMC cartridges, offering 16 nozzles and 10 pL drop volume. Furthermore, an industrial inkjet printer DMP-3000 from Fujifilm Dimatix was used to deposit the fundamental layers of the devices that were manufactured on a relatively large scale. For upscaling the process in the S2S manufacturing platform, industrialized SE3 printheads were employed, possessing 128 nozzles and a 35 pL nominal drop volume. The selection of these specific printheads was based on their expected reliability, robustness, runnability, compatibility, and ability to deposit ink with a high drop volume. The nozzle plate of the printhead also contains a special coating that allows a preferable interaction between the printhead and the ink, but simultaneously avoids flooding, thus ensuring superior jetting characteristics. The standard operating procedure for preparing the inkjet system for the printing process is proceeded with ink preparation that includes the following steps: intermixing of the existing components in the ink by physically shaking for some time, followed by ultra-sonification, and finally filtration by using appropriate resolution chemical filters to remove the inhomogeneity in the particle distribution inside the ink. Once the ink is ready, then the cartridge is filled in and the printhead is then attached to the cartridge, which then allows the initiation of the printing process. In case there is a requirement to clean and recover the nozzles, the printhead is purged with an overpressure to let the ink flow out and open up the nozzles that are either clogged or nonfunctional, thereby performing the cleaning cycle over an absorbing pad. An individually optimized and dedicated digital waveform was developed for individual ink that was loaded using the main window of the graphical user interface of the Dimatix Material Printer (DMP) software.

In Figure 3, the jetting waveforms for the different inks and deposition platforms are shown. In the figure, it can be clearly seen that the individual jetting waveforms developed for the printers DMP-2831 (DMC printhead) and DMP3000 (SE3 printhead) for the different inks were different, and for the same ink and different printheads were similar but not same. The jetting waveforms for the inks were designed independently according to corresponding viscosity, material contents, density, material loading, homogeneity, and surface tension of the ingredients within the ink. The designing of a jetting waveform was accomplished by varying and adapting the parameter of pulse duration and magnitude of the applied voltage to the piezoelectric actuators, which was furthermore lined up into numbers of waveform pulses and segments and their application. As depicted in the figure, the waveforms generated for the PEDOT:PSS and P3HT:PCBM ink were found to be more complex than the ones for ZnO and Ag. The reason comes directly from the inherent property of the implemented ink. It could also be observed that the most important parameters, applied voltage and pulse duration for DMP2831, stayed in the regime of 28 ± 4 V and 9.3 ± 4 µs, respectively, whereas the optimal parameters for the industrial inkjet printer DMP-3000 were found to be in the range of 115 ± 15 V and 33 ± 4 µs. The reason behind this difference primarily was due to the difference in the offered drop volume from the printhead and the driving electronic and mechanical architecture and construction of the actuator inside the printhead. Meanwhile, the other deposition settings and digital pattern were fed in the system, which redirected to the last step, skewing of the printhead to a calculated angle that would generate the required printing resolution (dpi) or drop space (μm). This skewing to an angle was adjusted manually with the help of the rotating scale surrounding the installed printhead for DMP-2831 and was performed automatically in the case of the DMP-3000. The printing process of all the layers to develop the OPV stack was accomplished using jetting frequencies up to 5 kHz and a negative meniscus pressure of about 11 mbar. To accomplish certain predefined conditions, multiple layers were deposited e.g., ZnO, P3HT:PCBM, and PEDOT:PSS, always printed on a “wet on wet” basis for the same material. The entire printing process for all the fundamental layers was performed at room temperature and a relative humidity of about 22 ± 2 °C and 30 ± 5%. All the other supporting deposition parameters can be found in Table 1.

### 2.4. Characterization

Once the deposition process was accomplished for the individual or stack of functional layers, the process of characterization was followed up on to understand the electrical characteristic of the fabricated OPV device and the topology of either the individual layers or the entire stack itself. It was of high interest to also know the corresponding thickness and overlap of each layers. For the purpose of evaluating the electrical properties of the OPV, a setup consisting of an AM1.5G light source was realized to simulate the sun’s irradiation (equivalent to 854 W/m^2^) on top of the printed device, along with an Agilent E4980A (Agilent Technologies, Santa Clara, CA, USA) precision digital LCR meter and a Keithley 2636A (Keithley Instruments, Cleveland, OH, USA) source meter coupled to a measurement probe station to check the principle operation of the diode characteristics. In contrast, the layer-related characteristics and surface topology of the printed layers were evaluated using an optical and laser microscope from Leica (Wetzlar, Germany) and Keyence (Osaka, Japan) and a Veeco Dektak 150 (Veeco Instruments, Inc., Plainview, NY, USA) surface profilometer.

## 3. Results and Discussions

### 3.1. Development and Characteristics of an OPV Stack Based on a Lab Scale 

Using the described parameters in Table 1, the bottom Ag electrodes were printed using one layer. In contrast to that, all the other remaining layers in the functional OPV stack were printed using two layers, to ensure a defect-free and homogenously closed layer. A particle filtration process was performed just before filling the inkjet cartridges, especially for the inks containing photoactive material like P3HT:PCBM, as well as for the (ETL) ZnO layer and the top PEDOT:PSS electrode layer. After the printing of each functional layer, the samples were taken for microscopic analysis to evaluate the layer-related properties of the printed layers. Figure 4A shows a photo of a group of OPVs entirely printed using inkjet technology based on laboratory scale DMP-2831 and different active areas. The microscopic images of the individual functional layers (exemplarily shown for active area 2.5 mm × 5 mm) stacked over each other is shown in Figure 4B–E. The red arrow and the yellow ring on the image show the inkjet printing direction and the sequence of functional layer deposition. 

As mentioned before, the bottom Ag electrode was printed using a silverjet DGP 40LT-15C (Advanced Nano Products Co., Ltd, Sejong, Korea) nanoparticle ink and 40 μm drop space (635 dpi). After completing the printing process, samples were cured using the printer’s substrate platen at 50 °C for 30 min and then transferred to the oven for sintering at 150 °C for 20 min. This intermediate drying step was found to be very essential in controlling the spreading of the ink over the printed area. Figure 4B shows the microscopic image of the printed Ag electrode based on the first pattern (shown in Figure 1B) with different active areas varying between 7.5 to 17.5 mm^2^. The optical characteristics of the printed Ag layer showed heavy ink spreading over the PEN film, with a tolerance of about 100 µm. Nevertheless, the microscopic images showed the desirable homogeneous and closed layers for the printed solid patterns. The corresponding topology and thickness of the printed Ag layer was measured using a Veeco Dektak 150 profilometer for the test square patterns having dimensions of 2 mm × 2 mm. To obtain reliable measurement values, every time five printed samples (squares) were considered for the individual functional inks. The linear measurement scans were performed three times along and across the printing direction. The measurements showed that the ink accumulated to certain locations, especially where the printing started. The average thickness of the printed Ag layers considering 40 µm drop space and both the printing directions was about 250 ± 50 nm, along with a characteristic coffee ring effect. The ETL layer was printed using ZnO N-11-Jet ink and 25 μm drop space (1016 dpi). The dimension of the pattern was fixed at 7.5 mm × 9 mm (67.5 mm^2^). Since the ink consisted of fast-evaporating solvents and additives, special care was taken to limit the rate of evaporation by limiting the substrate platen temperature to 28 °C. Two consecutive layers were deposited to ensure fully closed layers without any pinholes. After the printing process was accomplished, the samples were cured using a convection oven at 135 °C for 20 min and were then analyzed under an optical microscope. Figure 4C shows the microscopic image of two printed ZnO based ETL layers deposited wet on wet. The transparent-like behavior of the printed layer refracts the incident light showing rainbow colors in the captured image. This characteristic indicated the occurrence of an evaporation-driven drying process, resulting in fully closed, but slightly inhomogeneous layers. The layer topology of the printed ETL and the other functional layers were analyzed using the same profilometer. As explained before, squares having the same dimensions were inkjet-printed on the substrate (every functional layer) and were then utilized for the measurement process. The measurement process was also fixed to a standard procedure. The measurement indicated that the printed ZnO layers did not show any sudden peaks over the entire layer topology. It is anticipated that the main reason in this case could have been the implementation of the physical particle filtration process (suggested by the ink supplier), restricted an uneven distribution of nanoparticles from entering the execution of the printing process. Finally, the average thickness of the ZnO based ETL layers was calculated and was found to be 200 ± 50 nm. The photoactive layer was printed using P3HT:PCBM ink and 25 μm drop space (1016 dpi). A physical process contributing to particle filtration (through a 5 µm PTFE filter) was employed and found to be extremely essential for the deposition of the ink, just before filling it into the cartridge, to avoid any undissolved polymer particles deposition over the printed layer. As mentioned earlier, the dimension for these patterns was fixed for all the different electrodes to 7.5 mm × 9 mm (67.5 mm^2^). Here, the ink for the photoactive layer was deposited using two layers. After the printing process was accomplished, the samples were cured using a convection oven at 120 °C for 20 min. Figure 4D shows the microscopic image (corner location) of a printed P3HT:PCBM layer over the already deposited ZnO and Ag layer. From the image, it can be seen that the printed layer formed a concentric ring shape-like contour at the center location of the active area, which was basically the result of the evaporation process from the 1,2-dichlorobenzene solvent. In order to counteract this phenomenon, the substrate or printer’s platen temperature was set to about 50 °C, to enhance the rate of evaporation. Nevertheless, the layer was found to be fairly thick, with full material coverage, and optically it exhibited a dark orange (wet condition) to purple (dried condition) appearance. The photoactive layer was analyzed for morphology for the test square patterns, which was printed using the same print settings. It was recognized that the value for the layer thickness varied from one part to another of the printed layer in the range of 100 nm, whereas the thickest part was settled at the print starting location. The evaporation of solvent during the printing process affects the layer morphology drastically by forming a concentric ring-like shape throughout the printed area. Therefore, it was seen that the printed layers showed relatively inhomogeneous topology, with recurring peaks arising at regular intervals. In order to avoid these abrupt peaks in the layer, which happened due to undesirable particle aggregation and inappropriate solvent evaporation, multiple filtration steps and an elevated substrate temperature were employed during the ink preparation phase and printing process. The average thickness of the layer was found to be 150 ± 50 nm. For the OPVs based on the conventional architecture, the conductivity of the anode electrode played a vital role in the collection of the holes and the functionality of the device. Thus, the top PEDOT:PSS anode electrode layer was printed using two layers to obtain a relatively thick layer. The chosen drop spaces for depositing the two layers were 15 μm (1693 dpi) and 11 μm (2309 dpi). After the printing process was accomplished, the printed samples were dried and cured using a convection oven at 100 °C for 20 min. The curing technique at this process step is considered very sensitive, and that is why it was performed at a lower temperature. This is primarily done to just drive out the water content from the PEDOT:PSS layer, and secondly to avoid any damage to the understacked layers. Figure 4E shows a microscopic image (location of the active area) of the top electrode. From the image it was observed that the spreading of the PEDOT:PSS ink over the P3HT:PCBM layer was relatively low (about 100 μm). Nevertheless, the printed layers showed high homogeneity with full material coverage and overlapping on top of the bottom Ag electrode. The layer topology of the PEDOT:PSS top electrode was analyzed using the test square patterns. As mentioned previously, the PEDOT:PSS electrode was printed with two layers using two different drop spaces, the first layer with 15 µm drop space and the second with 11 µm drop space. In order to collect detailed data about the layer topology, a further set of samples was printed. The first sample set was printed using only one layer with 15 µm drop space, and the second set of samples was printed using one layer with 11 µm drop space. It was recognized that the printed layers showed a uniformly distributed flat surface of the printed layer that was not recognized by any of the other used inks. A difference in the layer characteristics was observed when the printed layer across the printing direction was considered. In order to achieve an undisturbed layer deposition of PEDOT:PSS material from the inkjet nozzles, in-situ cleaning cycles (purge and blot) were adapted. The time interval between the cleaning cycles during the printing process divided the printed area into several stripes. Each stripe had its own layer morphology. The maximum and minimum thicknesses of the printed layer using 11 µm and 15 µm drop spaces were found to be about 335 nm and 185 nm, respectively. Whereas, the average thickness of the layers considering both the printing directions was found to be 210 ± 25 nm and 290 ± 35 nm for the 15 µm and 11 µm drop spaces, respectively. In total, 14 devices were printed, and each sample had three different device sizes, which meant the measurement of resistance was performed for 42 devices using a digital multimeter. The average resistance value for most of the devices was in the range of megaohms, which meant that there were no short circuits in the devices between anode and cathode electrodes and that theoretically the chosen architecture and material stack would lead to a successful OPV fabrication process.

### 3.2. Development and Characteristics of an OPV Stack Based on a Fab Scale

As a next step, the OPV devices of similar architecture and material stack were printed using the industrial inkjet printer DMP-3000, in order to test the possibility of printing the devices in a large scale and to measure their range of performance and efficiency regime. The deposition parameters were adjusted to fit the fabrication process requirements i.e., SE3 printheads with 128 nozzles and a drop volume of 35 pL. The layout was designed by printing several devices filling a complete DIN A4-sized sheet. As depicted in Figure 2, the sheet was divided into six separate arrays with different device sizes. Each array contained 36 patterns repeating the same device size e.g., Array 1 with 1 mm × 5 mm (5 mm^2^), Array 2 with 1.5 mm × 5 mm (7.5 mm^2^), Array 3 with 2 mm × 5 mm (10 mm^2^), Array 4 with 2.5 mm × 5 mm (12.5 mm^2^), Array 5 with 3.5 mm × 5 mm (17.5 mm^2^), and Array 6 with 4 mm × 5 mm (20 mm^2^). The bottom Ag electrode was printed using silverjet DGP 40LT-15C nanoparticle ink from Sigma Aldrich. The entire set of 216 devices with six different active areas with the defined A4 sheet were printed using 65 μm drop space (391 dpi). It is quite straight forward to understand the implementation of a lower resolution digital print pattern, which was basically due to the usage of a high drop volume, resulting in an escalated drop spreading and early drop coalescence. After the printing process was accomplished using the most optimal drop ejection parameters, the samples were sintered using a vacuum oven at 135 °C for 30 min. The second in the layer sequence was the ETL layer, which was printed in the size of 7.5 mm × 9 mm two times on top of each other over the Ag bottom electrode, without curing in between (wet on wet). The deposition of homogeneous pinhole-free layers was ensured using the ZnO nanoparticle-based N11 ink from Nanograde and print resolution of 40 μm drop space (635 dpi). The third layer, the photoactive P3HT:PCBM layer, was deposited also using the same dimension of 7.5 mm × 9 mm. The pattern was printed two times over each other (wet on wet), achieving a homogeneous layer using the formulated P3HT:PCBM ink and 40 μm drop space (635 dpi) with the material from Osilla Ltd. The top electrode layer was printed using PEDOT:PSS (CLEVIOS™ F HC Solar) ink from Heraeus. The print resolution of 35 μm drop space (725 dpi) with a total number of up to two layers was implemented to get a homogeneous thick conductive layer as the top anode electrode to achieve better layer conductivity and hole extraction. The choice of a low drop space value was determined for PEDOT:PSS, with the intention of depositing a maximum amount of material and thereby obtaining a fully closed layer. For instance, Figure 5A shows a photograph of a printed sheet using a DMP-3000 inkjet industrial printing machine. Each layer of the printed devices was deposited in one print run, filling the A4 sheet, and each section of the six sections contained 36 devices of the same size. Hence it was possible to obtain 216 devices within one print run.

The samples were taken for microscopic analysis for the optical characterization of the printed layers. The microscopic images of all the functional layers showed similar characteristics, as displayed previously in Figure 4B–E. The printed Ag electrode layer showed a homogeneously closed layer with sharp edges and an overspreading of 120 ± 20 µm. However, such a property from a printed feature was always expected due to the implementation of a high drop volume and print resolution (35 pL and a 65 μm drop space). When the optical analysis was done at the layer crossover of the incomplete device (without the top electrode), ring-like contours or structures were found to occur in the lower part of the printed active area. The reason behind this phenomenon could be the progression of slow evaporation rates of the solvent with regard to the high deposition rate, which can lead to slight material agglomeration. On the other hand, the printed PEDOT:PSS layer was found to be fully covered and homogenously distributed. The layer topology of the Ag electrode was analyzed using the Veeco Dektak 150 profilometer. The measurement was done from the actual printed sheets on the PEN film. The printed connection line between the active area and contact pad was chosen to perform the measurements. The measured distance was set to 1.2 mm in order to ensure a starting point from the PEN substrate and ending at the substrate again, after going through the printed Ag layer. The measurements were taken three times along the print direction for one line as well as three times across the print direction for another line. The measurements were performed at three locations to get an average thickness of the printed layers. The measurement across the printing direction showed that the ink was printed with a constant layer thickness at the middle location of the printed layer. However, the thickness of the layer was found to be slightly higher at the starting position of printing than at the end of the printing process, which could be explained by the deposition property of inkjet technology. The maximum measured layer thickness value was found to be less than 600 nm. On the other hand, the measurement along the printing direction showed the influence of the well-known coffee ring effect, as the layer thickness was found to be higher at the graph edges than at the middle location of the printed layer. Figure 5B shows the average thickness of the layers in a range of 450 ± 100 nm, when both the printing directions were considered. The layer topology for all the other functional layers was individually measured and analyzed using the same equipment and methodology, but using different sample locations and layers. The measurement for the ETL layer was done exemplarily for one layer. The measurement showed that the maximum layer thickness was about or less than 650 nm. The printed layer exhibited a coffee ring effect, and therefore the layer was found not smooth, as it showed rough surfaces along the measurement. The average thickness of the layer was about 180 ± 50 nm. To understand the actually deposited layers, another set of measurements was performed for the printed samples containing two layers, but deposited using the same print resolution. The influence of the coffee ring effect was also recognized in this sample. It is also worth mentioning that the layer topology at the middle location was found to be smoother and flat in the case of printing two layers. The graph related to the surface profile is shown in Figure 5C, which indicates an average layer thickness of 350 ± 80 nm. As mentioned previously, the photoactive semiconductor layer was printed with two layers of P3HT:PCBM ink with 40 µm drop space. The measurement was done initially for the printed sample for one layer with 40 µm drop space, and afterwards also for two layers using the same printing parameters. The maximum thickness for the printed layer was found to be approximately less than 500 nm. Meanwhile, the average thickness of the layer was about 210 ± 20 nm, as shown in Figure 5D. The layer thickness was observed to vary from one part to another, according to the accumulation of the deposited ink. Overall, from the measurement it was found that the layer was thicker at the beginning of the printing process, when compared to the latter part of the printing process. As mentioned before, it was also recognized that the deposited ink formed ring-shape contours and structures, which affected the layer thickness and homogeneity proportionally. In contrast, the top electrode layer was printed with three layers of PEDOT:PSS ink with 35 µm drop space for the final device. The maximum layer thickness for the printed layer was found to be just less than 600 nm. The ink was seen to accumulate at the center location of the printed layer and showed a high value among other printed layers i.e., an average thickness of about 450 ± 50 nm, as shown in Figure 5E.

In contrast to the topological measurements performed for the individually printed functional layers on the substrate, the actual buildup of the OPV stack was also characterized. The main goal of this comparative study was to evaluate and validate the topology of the printed functional layers when they were deposited either on the same surface or different surfaces. The profiles of the printed ZnO layer on the Ag electrode; P3HT:PCBM on Ag/ZnO; and a PEDOT:PSS electrode on Ag/ZnO/P3HT:PCBM can be seen in Figure 5F. As mentioned earlier, the printed Ag layer had a thickness of about 450 nm, with a fairly high tolerance of 100 nm. When two layers of ZnO were deposited on top of the Ag layer, the thickness of the stack reached 500 ± 50 nm, which indicated a different spreading behavior of the ink over contrasting surfaces of PEN and the Ag electrode. The values corresponding to the thickness of the measured layer proved a definite increment in the OPV stack. The high tolerance exhibited by the Ag layer as a result of the coffee ring effect was now reduced considerably. This is mainly because of the fairly dense distribution of ZnO ink material over the valleys of the Ag electrode. When the stack containing Ag/ZnO/P3HT:PCBM was considered, then the stack thickness was found to increase up to 750 ± 100 nm. This huge tolerance in the thickness value deviation was found to occur due to the inconsistent evaporation-driven pinning steps, yielding to the coffee ring contours (especially for the oDCB solvent) with respect to the rate of the deposition process. This same effect could also be seen when the printed P3HT:PCBM layers were characterized using laser scanning microscopy within the Appendix A. Due to the occurrence of the intensified coffee ring effect, waves were formed with height of about 300 nm, occurring at every 150 µm. In comparison, the exterior boundaries of the printed layer exhibited peak heights of up to 1 µm. When the entire OPV stack was accomplished, including the PEDOT:PSS electrode layer on top of the previously deposited Ag/ZnO/P3HT:PCBM stack, the device thickness was found to reach 1.4 ± 0.3 µm. The huge surface-related topological deviation for the printed PEDOT:PSS layer over the P3HT:PCBM layer was found to mainly occur due to the difference in the filling and correction of the topological contours from the understacked layers, and secondly due to the selective positioning of the PEDOT:PSS layer over the active area and not entirely over the BHJ layer.

### 3.3. Electrical Characterization

The samples were taken to the resistance measurement after printing all the layers. A digital multimeter was used to measure the resistance for all the printed OPV devices. As mentioned before, the devices were divided into six groups according to the size of their active areas, and each group consisted of 36 devices having the same size of the device active area. The average resistance value for each group of devices was not only affected by the varied device sizes, but also by the printed layer quality, the thickness, and its homogeneity. The measurement showed an average resistance of each individual area in the range of MΩ, which indicated there were no short circuits and that the devices were now ready to be characterized as OPVs. After completion of the initial characterization for the devices, the printed sheets containing the OPVs were evaluated for current versus voltage (I-V) characteristics, to figure out the range of performance and efficiency of the devices. In Figure 6, an example of a current versus voltage (I-V) characteristic curve for the inkjet-printed OPVs is shown that includes the active area corresponding to 17.5 mm^2^ and seven randomly chosen functional devices in total. The measurement was performed in darkness and under illumination (854 W/m^2^). From the graph, it can be seen clearly that the devices showed a typical diode characteristic curve under darkness, where they showed blocking behavior in the reverse bias or negative regime. At about +0.3 V the diode switched ON, displaying the forward current. On the other hand, under illumination the devices showed OPV characteristics, where they started to show negative photocurrents of about −0.35 mA/cm^2^ in the third quadrant, along with a voltage of about −0.5 V. As the illumination process proceeded, the OPV characteristic was enabled, which mainly occurred in the fourth quadrant of the graph where the short circuit current density (*I_s_*_c_) and open circuit voltage (*V_oc_*) reached its utilizable maximum power point (Mpp), with *I_sc_* and *V_oc_* values of −0.2 mA/cm^2^ and + 0.25 V, respectively, yielding to the point of maximum power density of about 0.05 mW/cm^2^. 

With regard to the values obtained from the manufactured devices, it could be interpreted that the performance of the devices was not optimal when the electrical performance of the OPVs were compared to ones exhibited in the literature that were partially printed e.g., a *V_oc_* of 0.89 V, an *I_sc_* of 9.98 mA/cm^2^, a FF of 56.78%, and a PCE of 5% [14] and printed e.g., a *V_oc_* of 0.59 V, an *I_sc_* of 3.64 mA/cm^2^, a FF of 37%, and a PCE of 1.31% [36]. On the other hand, it should be noted that the present devices were manufactured and measured in the ambient atmosphere, without any encapsulation. The OPVs (ITO-free) were furthermore intentionally manufactured using a known low performance organic BHJ photoactive material because of its solution processing robustness. Measurements were performed for all the manufactured OPVs (216 devices), which had active areas varying from Array 1 with 5 mm^2^, Array 2 with 7.5 mm^2^, Array 3 with 10 mm^2^, Array 4 with 12.5 mm^2^, Array 5 with 17.5 mm^2^, and finally to Array 6 with 20 mm^2^. In the histogram shown in Figure 7, the statistics of the dependence for *V_oc_*, *I_sc_*, fill factor (FF), power conversion efficiency (PCE), as well as the manufacturing yield depending on the variation of the active area for the devices can be seen. It can be seen in Figure 7A that the *V_oc_* increased slightly from 0.38 ± 0.03 V to 0.42 ± 0.06 V, when the size of the devices was increased from 5 mm^2^ to 20 mm^2^. It can be concluded that the *V_oc_* for the devices tended to reach the competitive goal of about 0.4 ± 0.2 V. When the *I_sc_* was evaluated, it was found in Figure 7B that the current density increased steadily from 0.17 ± 0.04 mA/cm^2^ to 0.28 ± 0.05 mA/cm^2^.

In comparison, when the dependency of FF was considered, it was found (Figure 7C) to have been approximately constant at around 34 ± 3.2%. Whereas, the same could not be said to be true for the property of PCE (Figure 7D), where the values ranged from 0.02% up to 0.18% when the size of the devices were upscaled from 5 mm^2^ to 20 mm^2^. These two indicators showed that the organic BHJ material P3HT:PCBM was definitely limited in the performance under ambient atmosphere, where the deposition and measurement process were performed. On the contrary, it can be discussed within this research paper that the devices that are manufactured using industrialized inkjet technology offers electrical characteristics with <20% tolerance. This value can be considered as low when they are compared with state of the art publications about printed OPVs. There are in general few publications that focus their research on the topic of statistics and definitions of electrical and manufacturing tolerances for printed OPVs. Next to this, here the most important statistical aspect was also measured—the manufacturing reliability or yield of the inkjet-printed OPV arrays. Under ambient conditions, it could be expected that the number of quantified defects would increase, with increasing printed area. The same interpretation can also be seen in Figure 7E, where the manufacturing yield for the printed OPVs tended to decrease from about 98% down to 65% when the size of the devices increased from 5 mm^2^ to 20 mm^2^, which has also been explained in the literature for other devices using the same printing platform [45]. It was mentioned in the section before that the printable area for ZnO and P3HT:PCBM was always kept constant, whereas it was varied for the Ag cathode and the PEDOT:PSS anode layers. It is interpreted that the defects might primarily have been caused during the printing process due to the evaporation-driven drying process (resulting in high layer roughness and steps- or wave-shaped contours) for the ZnO and P3HT:PCBM layers (shown in Appendix A and explained also in the literature) [14]. And finally the electrical short circuits (low terminal resistance) could be detected when the top PEDOT:PSS anode electrode came in contact with the defect prone zone over the photoactive BHJ layer. Although the observed manufacturing yield of the printed OPV was about 70%, the reliability could definitely be increased by improving the layer homogeneity and utilizing a combination of solvents that can dissolve P3HT:PCBM within the ink formulation. Alternatively, the layer morphology could also be improved by varying the rate of deposition, by applying a lower drop volume and a higher substrate temperature. The development of different batches containing inkjet-printed OPVs and arrays led therefore to the definition of manufacturing tolerance, which in this case was about 5%. The performance of the printed OPVs could be improved by following two routes: Improving the manufacturing process and adapting the electrical characterization. In the case of the manufacturing process, the deposition of functional layers could be accomplished in one single batch, with an implementation of new BHJ and charge transport materials with suitable ink formulations for inkjet technology. On the other hand, it could also be expected that the energy levels of the functional materials and layer interfaces could vary due to doping, upon the interaction of the ambient atmosphere. This could be prevented by encapsulating the devices or performing the electrical characterization in an inert atmosphere.

## 4. Summary

Within this research paper, it can be summarized that it is possible to upscale the manufacturing process for printed OPVs from laboratory to an industrial scale using inkjet technology consisting of equipment and accessories from Fujifilm Dimatix over the S2S printing platform and area of DIN A4. It is worth mentioning that the developed OPV stacks were manufactured on standard flexible plastic substrate, under ambient atmosphere, and free of conservative materials such as ITO. In order to accomplish an appropriate manufacturing process, a conventional device architecture was selected with a simplified layer sequence of Ag/ZnO/P3HT:PCBM/PEDOT:PSS. The physical geometry of the functional layers, the corresponding printing process, and the related post-treatment parameters were systematically tuned to achieve the most optimal layer stack for the printed OPVs. Here, the scalability of the manufacturing process was also demonstrated by developing the inkjet-printed OPVs in the form of arrays that contained device quantities up to 36 in one single array and up to 216 devices in one single DIN A4-sized plastic sheet containing six arrays. The size of the devices was upscaled by varying the active area of the OPV devices, for example from 5 mm^2^ to 20 mm^2^. By following this route of investigation, the tolerances for the manufacturing yield and device electrical performance (using inkjet technology) could be defined, which was in the range of 10% to 20%. The most optimal results showed that all the inkjet-printed OPVs developed within this S2S-based industrial scale manufacturing process with a set of specific functional materials offered electrical characteristics of: *V_oc_* 0.48 V, *I_sc_* 0.33 mA/cm^2^, FF 37.2%, PCE 0.18%, and manufacturing yield more than 70%. 

## Figures and Tables

**Figure 1 micromachines-09-00642-f001:**
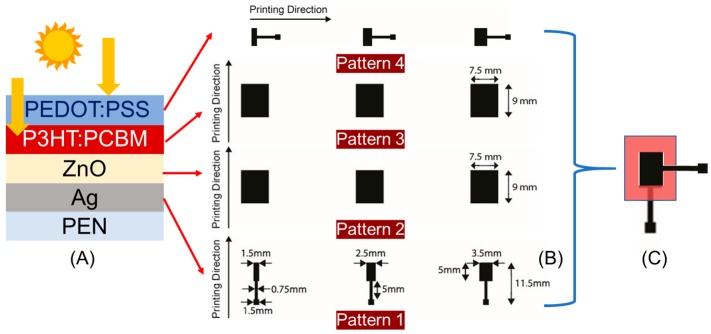
Image showing a (**A**) conventional OPV device architecture, (**B**) an overview of the designed digital patterns 1–4 with bottom Ag and top PEDOT:PSS electrode layers with different sizes and constant size of the electron transport layer (ETL) and bulk heterojunction (BHJ) layers, and (**C**) a top view of the intended OPV device.

**Figure 2 micromachines-09-00642-f002:**
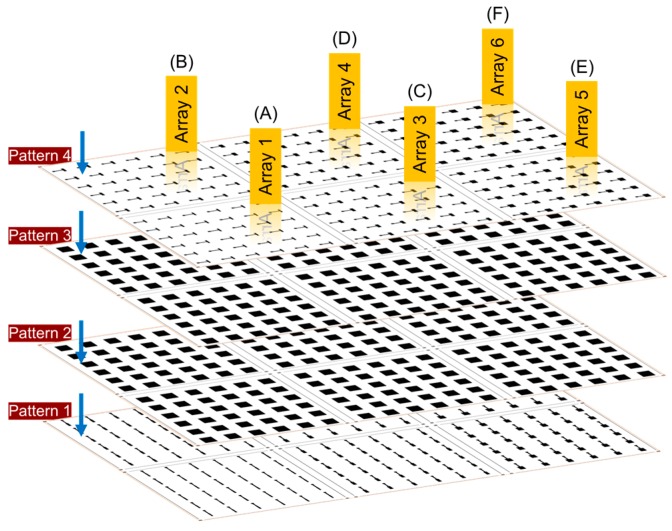
Digital pattern of the Ag bottom electrode layer for the different areas: (**A**) 5 mm^2^, (**B**) 7.5 mm^2^, (**C**) 10 mm^2^, (**D**) 12.5 mm^2^, (**E**) 17.5 mm^2^, (**F**) 20 mm^2^.

**Figure 3 micromachines-09-00642-f003:**
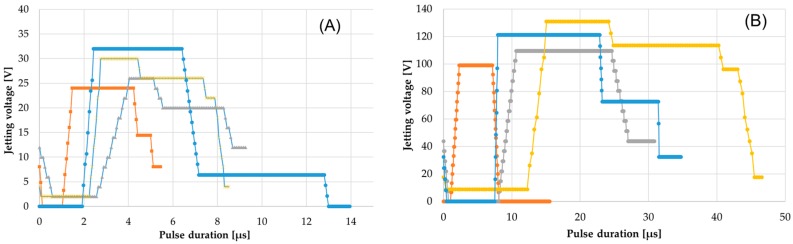
Images exhibiting the suitable jetting waveforms for the different inks implemented for depositing the functional layers of the OPV devices, using (**A**) DMP-2831 with DMC printheads; and (**B**) DMP-3000 with SE3 printheads.

**Figure 4 micromachines-09-00642-f004:**
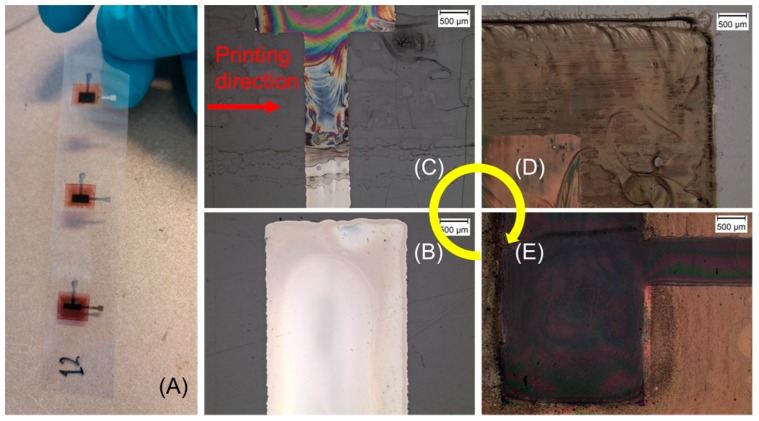
(**A**) Photo of all inkjet-printed OPVs of different active areas and microscopic images of the printed functional layer corresponding to (**B**) the bottom Ag cathode electrode, (**C**) the ZnO ETL layer, (**D**) the P3HT:PCBM bulk heterojunction or photoactive layer, and (**E**) the top PEDOT:PSS anode electrode layer, shown exemplarily for 2.5 mm × 5 mm.

**Figure 5 micromachines-09-00642-f005:**
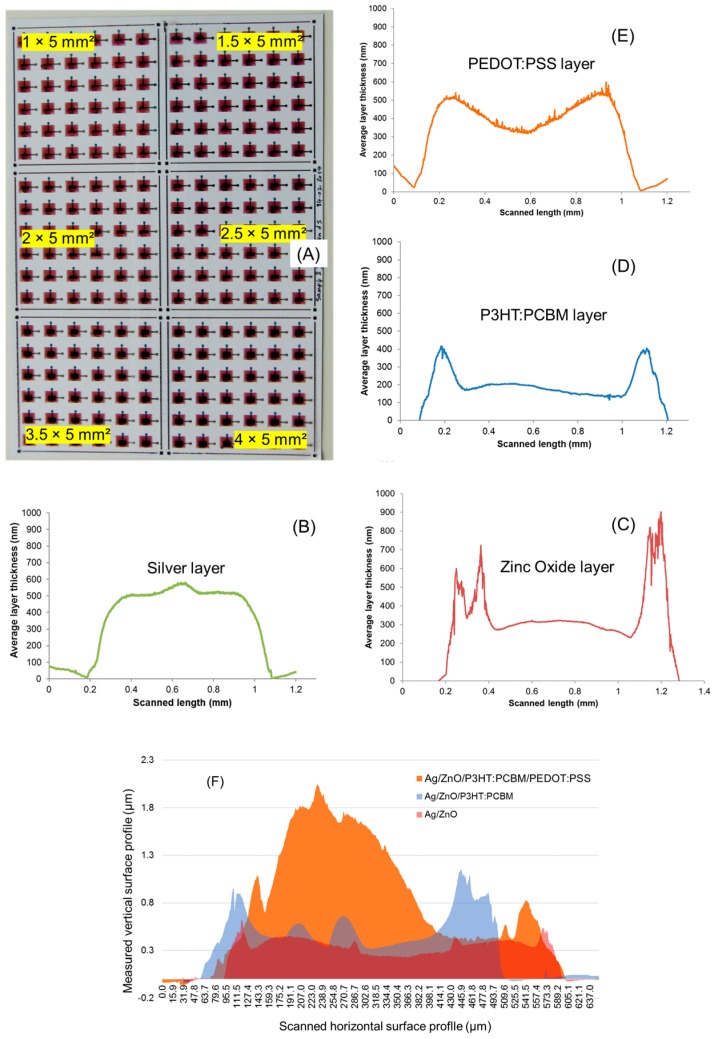
(**A**) Photograph showing OPV array with all six active areas printed using the DMP-3000 i.e., 5 mm^2^, 7.5 mm^2^, 10 mm^2^, 12.5 mm^2^, 17.5 mm^2^, and 20 mm^2^; (**B**–**E**) average surface profiles of printed layers for Ag bottom electrode deposited using silverjet DGP LT-15C nanoparticle-based ink with 65 µm drop space, two electron transport layers using ZnO (N11) ink with 40 µm drop space, an active layer using P3HT:PCBM ink with 40 µm drop space, and a top electrode layer using PEDOT:PSS CLEVIOS™ F HC Solar ink with 35 µm drop space. In comparison, (**F**) shows the buildup of the actual OPV stack using the same deposition parameters.

**Figure 6 micromachines-09-00642-f006:**
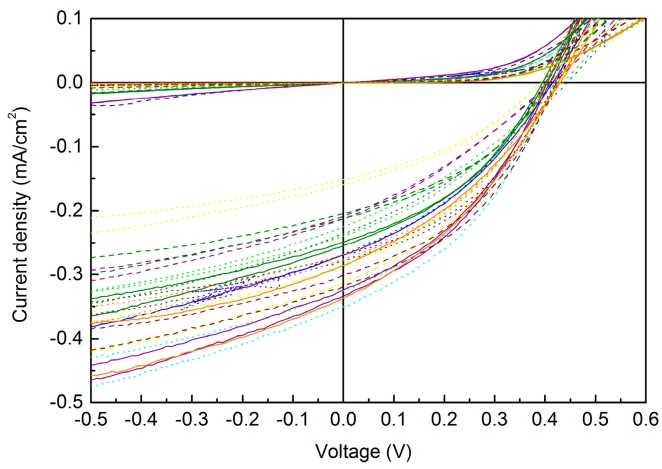
Graph showing the current versus voltage electrical characteristics of the all inkjet-printed OPVs with 17.5 mm^2^ as an active area.

**Figure 7 micromachines-09-00642-f007:**
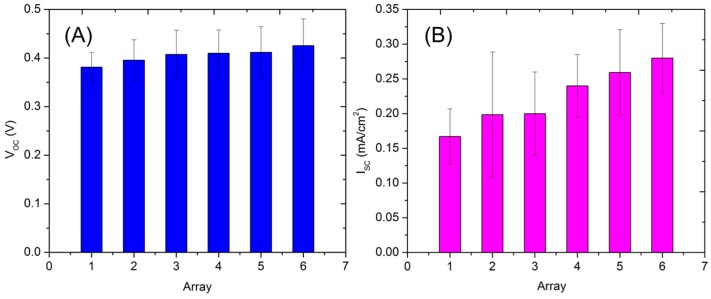
Graphs showing the statistics behind (**A**–**D**) the electrical properties and (**E**) the manufacturing yield of the all inkjet-printed OPV device arrays having 216 devices, with Arrays 1–6 having active areas of 5 mm^2^, 7.5 mm^2^, 10 mm^2^, 12.5 mm^2^, 17.5 mm^2^, and 20 mm^2^, respectively.

**Table 1 micromachines-09-00642-t001:** The deposition parameters optimized for the development of functional layers dedicated to the manufacturing of OPV devices.

Functional Layer	No. of Nozzles	Jetting Voltage (V)	Substrate Temperature (°C)	Cartridge Temperature (°C)	Drop Space (µm) & No. of Layers	Curing (°C, min)
**DMP-2831 & DMC Cartridges, 10 pL drop volume**						
1. Ag	12	24	28	36	40 & 1	150, 30
2. ZnO	8	26	28	36	25 & 2	135, 20
3. P3HT:PCBM	10	30	60	50	25 & 2	120, 20
4. PEDOT:PSS	6	32	28	28	15, 11 & 2	100, 20
**DMP-3000 & SE3 printheads, 35 pL drop volume**						
1. Ag	40	99	45	28	65 & 1	150, 30
2. ZnO	36	110	55	28	40 & 2	135, 20
3. P3HT:PCBM	50	131	70	50	40 & 1	120, 20
4. PEDOT:PSS	30	121	28	28	35 & 2	100, 20

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
