# Peer review of "Manufacturing of All Inkjet-Printed Organic Photovoltaic Cell Arrays and Evaluating Their Suitability for Flexible Electronics"

_micromachines, 2018, doi:10.3390/mi9120642_

Round 1

Reviewer 1 Report

Journal: micromachines

Article Title: Manufacturing of all inkjet-printed organic photovoltaic cell arrays and evaluating its suitability for flexible electronics 

Authors: Kalyan Yoti Mitra, Abdelrahman Alalawe, Stefanie Kreissl, Christine Boeffel, Reinhard R. Baumann 

Reviewer comments:

In this work, Kalyan et. al. fabricated organic photovoltaic arrays with all inkjet printing technology, and evaluated the optoelectronic properties of these printed cells. The authors really give a detailed procedure from device fabrication to characterization. Considering that most of papers on OPV is based on a spin-coating processing, I will try to encourage publication of this work. Because commercialization is the only future for OPV, although the physical process from light absorbing to power generation is of particular academic interest. However, I will encourage the authors to provide more insight into the printing technology they adopted:

1. Device failure analysis should be performed. I suggest the authors to compare their result with literature report, and find the failure mechanisms regarding to the lower efficiency compared to the spin-coating process.

2. Try to compare this technology with other deposition methods, and show the advantages and a possible promising future within this technology.

3. I can not understand the purpose to fabricate small area arrays on a large sheet. Because that will cause considerable transmission loss due to the effective area for energy conversion.

Author Response

Please find attached the document containing the response from us.

Reviewer 2 Report

You have presented a detailed technology report of a study involving arrays of fully printed OPV stacks under ambient conditions. The paper provides complete experimental detail which is valuable to future studies. 

Please perform the following minor revisions prior to publication:

1) Pg2ln3: PPS should be PSS

2) Pg6ln23; cite the source for the ZnO work function and specify the conditions (film thickness)

3) Pg6ln26; specify if the 12.5 mg/ml is total solid content or for each component

4) Pg8; section 2.3; specify the light source and explain how the radiation spectrum is different form AM1.5G

5) Fig 4: red text is blurry, please update; also layout is counter intuitive; please modify

6) pg10ln34; substrate temperature countradicts TABLE 1: please correct

7) Fig 5; please add print layer text label to each profile for simplicity

8) section 3.3; your PV effect is very poor; Can you please compare your printed device with a spin coated reference device?  Either from your lab (preferred) or from the literature with your chosen device architecture.

9) please include a band level figure of the work function and HOMO-LUMO levels of your printed layers. Explain which methods were used to determine the energy level values.  Use this graph to help explain your results

Author Response

(The authors gave the same response as above.)

Reviewer 3 Report

This study uses inkjet technology to fabricate an array of organic solar cells and its manufacturabililty. The reviewer recommends this pape to be published after answering questions below.

- The authors claim that the inkjet-printed layers are pinhole-free. The reviewer recommends to measure topology of each surface by AFM to prove their claim.

- The authors show the profiles of each printed layers in Figure 5. But, these do not show the actual profile when formed as a solar cell which is made of stacking the layers. 

- They have to show the ZnO profile on top of Ag layer, P3HT:PCBM profile on top of the Zn:Ag layer, the profile of PEDOT:PSS layer on top of the previous layer, rather than separate measurements. The profiles in the fabricated devices will be different from what they've measured

-The PCEs of printed solar cells are significantly low compared to other studies. What the authors think the reason is. Is there any solution to improve it. 

Author Response

(The authors gave the same response as above.)

Round 2

Reviewer 1 Report

I think this paper can be published as it is.

Author Response

Thank you!

Reviewer 2 Report

section 3.3; your PV effect is very poor; Can you please compare your printed device with a spin coated reference device?  Either from your lab (preferred) or from the literature with your chosen device architecture.

Response by authors – The comparison is already made in the introduction chapter, but now is explained again with details in Results & discussion chapter in pp. 15, line no. 33 to pp. 16, line no. 1. “With regard to the values obtained from the manufactured devices, it could be interpreted that the performance of the devices is not optimal, when the electrical performance of the OPVs are compared to ones exhibited in literature which is partially printed e.g. Voc 0.89 V, Isc 9.98 mA/cm2, FF 56.78 % and PCE 5 % [14].”

-- You have not explicitly convinced the reader that your device architecture is suitable for OPVs. Also, the comparison you list here is misleading. In the conclusion, you have compared your results to a glass/PEDOT:PSS/PCDTBT:PC70BM/ZnO/Ag stack.  Please compare your printed results to a suitable reference cell that has the same architecture that you are using.  If not from literature than from your lab (e.g. spin coated).   Are you able to provide device I-V characterization under ambient and inert conditions?

Please include a band level figure of the work function and HOMO-LUMO levels of your printed layers. Explain which methods were used to determine the energy level values. Use this graph to help explain your results.

Response by authors - The energy level diagram of the implemented material stack is mentioned in some published works already (a) Jung et al., Organic Solar Cells: All-Inkjet-Printed, All-Air-Processed Solar Cells, Advanced Energy Materials, 2014. (b)  Cherian et al., Fabrication of Organic Photo Detectors Using Inkjet Technology and Its Comparison to Conventional Deposition Processes, IEEE Sensors Journal, 2018. (c) Mitra et al., Work Function and Conductivity of Inkjet-Printed Silver Layers: Effect of Inks and Post-treatments, Journal of Electronic Materials, 2018. and measurement methodology has been using UPS and XPS. Additionally, a sentence (in pp. 4 and line no. 5 - 7) is added “The intended OPV architecture is developed based on the material’s energy band levels implemented already in literature [14, 43, 44].” to cross-reference to the citations.  

-- Good additions and reference to material energy levels, however, please provide an energy level diagram for your stack as this is not exactly shown in any of the references you have listed.  It would be beneficial to show that you have thought of the electron and hole transport mechanisms.  You can also try  to explain how these levels can change if characterized in ambient air (vs N2).  In your discussion/conclusion it would be beneficial to describe the necessary modification to your device architecture to improve PCE and Isc.

Author Response

Please refer to the document attached. The rebuttal is marked in blue color.

Reviewer 3 Report

I agree to accept this paper in its form.

Author Response

Thank you!